# "We Knew No One Else Had Our Back except Us": Recommendations for Creating an Accountability Care Framework with Sex Workers in Eastern Canada

Kathleen C. Sitter [1,*], Alison Grittner [1], Mica R. Pabia [1] and Heather Jarvis [2]

1   Faculty of Social Work, University of Calgary, Calgary, AB T2N 1N4, Canada
2   Independent Researcher, Halifax, NS B3K 5Z7, Canada
*   Correspondence: kcsitter@ucalgary.ca

**Abstract:** The authors report findings from a 15-month project that focused on the experiences of sex workers who live and work in an Eastern Canadian province. As part of a larger multi-phased study, 15 adults who identified as women, transgender, or non-binary, and received money or goods for sexual services, participated in photo-elicitation interviews. Drawing on a critical framing analysis, findings indicated supports—as identified and experienced by sex workers—encompassed three categories of care: self, community, and collective. These categories are described, with a particular focus on the latter two. Continuing with the care-based framework, recommendations to structure interventions draw on the role of accountability care in identifying how best to operationalize policies that promote health, well-being, and dignity of Canadian sex workers. The paper begins with a brief overview of the Canadian context and the role of supports. It follows with a discussion on the materials and methods and the results. It concludes with recommendations, limitations, and future considerations.

**Keywords:** sex work; human rights; supports; self-care; community care; collective care; accountability care

## 1. Introduction

Sex work involves the exchange of sexual services for money, goods, or other material components. Sex work is also conceptualized as existing across a spectrum (Lanau and Matolcsi 2022), in which consent, control, power, and place play critical roles in diverse experiences, particularly when intersected with identities and social locations. Overgeneralizing and framing sex workers as a unified political identity erases the diversity of experiences across this spectrum, as sex workers' lives are multidimensional and complex (Desyllas 2014). Treating sex workers as a unified political identity discredits individual agency and intersectionalities across gender, age, race, ability, location, class, and access to power (Kunimoto 2018). This false sense of homogeneity results in policies and systems that fail to recognize the breadth and depth of experiences in this field (McBride et al. 2021) and ultimately do not meet the needs of sex workers (Grittner and Walsh 2020).

In the Canadian context, federal legislation follows the 'Nordic Model' in which sex work is deemed analogous to sexual exploitation, and the purchase (not selling) of sexual services is criminalized (Benoit et al. 2017). This creates a precarious lack of work policies for protection, leaving sex workers in a liminal criminalized space without a clear workplace safety net (Benoit et al. 2018). Consequently, sex workers' have scant access to resources, capacity to earn money, and ability to negotiate safe working conditions. The COVID-19 pandemic highlighted the support consequences of this policy quagmire when sex workers faced a loss of income, lacked access to government subsidies, and experienced higher-risk work environments without the same labour and legal protections granted for other professions. The Canadian Emergency Response Benefit (CERB) was intended to financially

support Canadians experiencing unemployment due to the pandemic yet required prior participation in the Canadian tax system, which is challenging for sex workers (Hensley and Bowden 2020). For sex workers who qualified for CERB, additional layers of stigma, judgement, and the fear of criminalization created barriers to accessing these government funds (Hensley and Bowden 2020). These barriers built upon pre-pandemic conditions whereby experiences of stigma prevented sex workers from accessing healthcare (Lazarus et al. 2012), social support (Duff et al. 2015), and securing other sources of employment (Begum et al. 2013).

For some sex workers, the risk of contracting COVID-19 was secondary to their inability to meet basic needs. Many were unable to transition their services to online platforms, and the loss of income was further compounded by absence of locations for in-person work and pressure to reduce service rates (Hensley and Bowden 2020; Mercer 2020). This vastly increased risks for sex workers by reducing agency and choice (Mercer 2020). Sex worker advocacy groups campaigned for timely and long-term government interventions, including no qualifying requirements for social supports, healthcare, and a universal income; ensuring sex workers have access to essential resources such as condoms and personal protective equipment; and access to safe and affordable housing (Shareck et al. 2021). These types of interventions require an understanding that uni-sector policies fail to address the overlapping identities and challenges commonly experienced by Canadian sex workers (Shareck et al. 2021). The ongoing COVID-19 pandemic has highlighted sex workers' need for multi-sectoral policy that crosses the domains of income, gender, housing, healthcare, and disability.

Sex workers exist in marginalized social, political, and economic positions (Zangger 2010). Government and institutions abdicate care for sex workers, perpetuating structural violence and stigma. Sex workers must provide care for themselves and their communities from the margins of society.

*Supports and Notions of Care*

As authors, scholars, and activists in this field, we consider the role of supports as being imbedded in notions of care and are inherently connected across micro- (the individual), mezzo- (group level), and macro-systems (structural). We use the term care to frame the nuances and complexities of supports across these levels. Hobart and Kneese (2020) observe that care is "theorized as an affective connective tissue between an inner self and an outer world, care constitutes a feeling with, rather than a feeling for, others" (2). Mobilized care provides, "visceral, material, and emotional heft to acts of preservation that span a breadth of localities: selves, communities, and social worlds" (Hobart and Kneese 2020, p. 2).

Self-care refers to acts of compassion towards oneself that are essential for preventing distress and burnout (Barnett and Cooper 2009). Bressi and Vaden (2017) instate how self-care promotes well-being by regulating the mind and body in such a way that the professional self has limited negative impacts on the personal self. They discuss strategies such as sleep hygiene, good nutrition, exercise, opportunities to be creative, rest, and community (Bressi and Vaden 2017). Godfrey et al. (2011) broadened this definition of self-care by defining care at both the individual and community levels. Community care involves a comprehensive web that encompasses other individuals, programs, and agencies (Godfrey et al. 2011). For example, Battle (2021) shares the powerful impacts of a beauty salon in Chesterfield County, Virginia, in supporting black girls navigating gendered and racialized structural violence in their communities. In this manner, a safe space for community care created opportunities to interrupt violence in the lives of these young women. Collective care augments this model by referring to supports at the systemic level, including policies, laws, and structures that disrupt and/or perpetuate oppression (Wilson and Richardson 2020). Wilson and Richardson (2020) further propose healing-centered collective care, a framework that expands on trauma-informed care, for marginalized communities amidst inequitable power relations. Ginwright (2018) describes healing-

centered collective care as a movement towards understanding healing as political rather than clinical. By conducting policy and legal analyses, the root causes of trauma are highlighted and mitigated and this provides a recipe for collective change that counters band-aid solutions for surface-level symptoms.

These nuanced understandings of supports and care can provide a helpful framework in understanding the experiences of sex workers, as well as understanding care as a "critical survival strategy" (Hobart and Kneese 2020, p. 2). Yet, despite the literature on the importance of self, community, and collective care in other populations (Battle 2021; Godfrey et al. 2011; Ginwright 2018), there is a paucity of research on holistic levels of care for sex workers specifically. Emerging evidence suggests care in the form of peer support is a strong mechanism for resisting stigma related to sex work. Huang's (2015) research with Australian sex workers identified peer support as a key mechanism for shifting sex work stigma and its deleterious impacts. As Ahmed (2014) writes concerning sex care in relationship to structural violence, "Some have to look after themselves because they are not looked after: their being is not cared for, supported, protected" (np). Understanding the role of care and support in a field with no structural safety nets, we must consider how this is experienced and understood by sex workers themselves. Recommendations must be grounded in the diversity of lived experiences within sex work communities (Ham and Gerard 2014). Addressing these gaps, our research explores how supports are experienced and understood by sex workers in an Eastern Canadian province.

## 2. Materials and Methods

This article reports on one area of a larger study in partnership with a community organization that involved exploring the social location of sex workers living and working in an Eastern Canadian province. As part of this study, 15 sex workers participated in photo-elicitation interviews. Subsequently, participants created a photograph and arts-based exhibit that was distributed through social media and place-based settings such as pharmacies, community hubs, and post-secondary institutions. The research was conducted in partnership with a community organization and received institutional Review Board approval.

### 2.1. Profile of Participants

Participants were recruited throughout an Eastern Canadian province via social media, community posters, local venues, and post-secondary institutions. A total of 15 adults who identified as women, transgender, or non-binary persons who received money or goods for sexual services and had been working and/or living in the province for no less than 6 months and up to 30 years participated in the study. All participants were adults, between the ages of 18 to 55, and living and working across the province. Many participants worked and lived in different locales over time: Of the 15 participants, several had or were working in rural locations (7), had or were living in rural locations (3), and had or were working in city-centres (13). While many had been or were independent entrepreneurs (11), many participants also laboured in various sectors with a range of current and previous experience in indoor work, including exotic dancing (2), online and various types of virtual interactions (3), adult massage studios (8), co-operatives (5), BDSM/Dom (4), and outdoor work, including street-based work (4).

### 2.2. Process

Participants were asked to take photographs of their life experiences as a person who engages in sex work, followed by an individual interview. Interviews took place during 2018–2019 and were conducted face-to-face at the community partnership organization, at a program location, or a public location chosen by the participant. Participants had the option of participating in two follow-up group meetings. In line with best practices, all participants received honorariums for the interviews and group meetings.

Preliminary analysis and emergent themes that guided the group process involved critical framing (Sitter 2015). This process recognizes that a participant's interpretation of their image carries the most power and importance of visual meaning (Stanczak 2007). When large amounts of data are present, researchers undertake analysis within the context of the other data and the overall theoretical framework (Guillemin and Drew 2010, p. 184). The preliminary themes were shared at the group meetings for feedback and discussion. The final group themes from this process included basic needs, laws, healthcare, the industry, and supports.

While key learning points from the group sessions have been documented elsewhere (see Sitter et al. 2022), this article provides a focused review on the photo-elicitation interview data. As part of the interview, participants used photographs to convey and represent experiences that affect their lives as a person living and/or working as a sex worker. Interview questions were based on image guidance, specifically how the images represented (1) life experiences as a person who engaged in sex work, (2) social and personal factors, (3) laws, (4) geography (work, travel, and home), and (5) needed supports. As a second form of analysis focused on the interview data, members of the research team revisited and reviewed all transcripts for a period of four months. Guided by the research questions, all interview data were reviewed and involved sustained engagement until clear themes emerged. This included the ways in which participants described their photographs in regards to providing further context to their experiences. We noticed that the theme "supports" was linked to each theme named above, providing a deep structure that framed the data. By conducting an in-depth analysis of the individual interviews, the idea of supports evolved to encompass aspects of care, which are used to organize the findings and recommendations. These four aspects are broadly defined as follows:

- <u>Self-care</u>: Involves the acts taken at an individual level to care for oneself;
- <u>Community care</u>: Describes a larger network of supports that, at times, is reciprocal. Includes individuals as well as programs and agencies;
- <u>Collective care:</u> Includes the systems and structures that create a ripple effect that support and/or hinder people engaged in sex work—legislation, policies, funding, and government services;
- <u>Accountability care</u>: The methods by which individuals, programs, and systems are held responsible and accountable. This accountability is necessary to build capacity for collective care and includes recommendations based on research findings.

The following section relies heavily on narrative quotations from participants. However, several participants indicated they did not want their names used. Therefore, to respect the wishes of these individuals, all names associated with indirect and direct quotes have been omitted, along with potential locations, images, or descriptions that could reveal the identity of an individual, including the names with narrative comments could threaten anonymity as readers familiar with the community might be able to glean the identity of a given speaker.

## 3. Results

Findings are situated across themes of care, recognizing that everyone has their own lived experiences and methods of supporting themselves or others across these domains. Results in this section focus on community care and collective care for several reasons. Self-care was discussed by participants, as "self-care is necessary for collective survival within a world that renders some lives more precarious than others" (Hobart and Kneese 2020, p. 5). Within Canada, the responsibility of care for sex workers largely rests on the individual and their immediate community. The policies and legal context in Canada leave individuals alone in managing risks; self-care and individual accountability comprise the bedrock of all levels of supports. However, looking toward future change generated by structural supports, community and collective care hold transformative strategies rooted in an accountability of care. The areas of community and collective care are described and divided into subthemes in this section.

*Community Care*

"I need some kind of support system."

Community care involves pursuing the well-being of oneself and others, where safety is viewed as a shared responsibility. Participants shared a number of themes describing the characteristics of community care: (1) collective empathy, (2) shared safety, and (3) available programs and services accessible in both larger centres and smaller communities.

*Collective Empathy.* A core aspect of community care included fostering collective empathy. Forming community with others who have similar work and life experiences was an integral part of collective empathy often found through online and face-to-face support groups in the community and beyond that are run by agencies and/or sex workers. Collective empathy does not mean a collective voice of shared experiences; rather, it includes the recognition of diversity in the field. To highlight this, one participant explained that she held a lot of privilege when it came to the industry, and yet she had difficulty accessing resources: "I may not have access to all of these things but for the vast majority of sex workers out there, they have even less access to these." Another individual explained how notions of choice are often very limited for many sex workers: " . . . survival sex work is not a freely made choice and I think that's something that we kind of miss over a lot. That, you know, survival sex work is a choice, but it's not a fully or freely made choice." Recognizing differences in social location and the inherent impact on choice, experiences, and access to resources is an important critical lens for community solidarity.

Across indoor and outdoor work, another quality of creating collective empathy—consistently noted as being extremely important for well-being—involved building community with other sex workers. One participant explained: "I have this amazing circle of women that just trust each other and support each other and want to see each other do so much better and . . . you have to take care of yourself, right?" Having a shared experience of sex work was important to this sense of support: "Just being around other people who know what work is like."

One participant reflected on her previous experience working at an adult massage studio, which overall was not an ideal work situation, but she missed the social aspects: "I made friends with the women I worked with in [location]. We would go out; we would go to [location] together. We would hang out. We became friends, um, and I still think about them today."

Drop-in programs led by support services providers were also an important place for creating these connections. Participants described the unique sense of community with other sex workers as an essential aspect of such programs: "I have no friends other than the girls in [name of program] . . . everyone looks at us like, well you know, 'you don't have to do this' but like I mentioned survival, you know? Half of [name of area] hasn't been through the shit that we've been through."

Peer support was a critical element of community support, allowing sex workers to relate to one another in a non-judgmental manner while also providing advice and guidance. Connections took place face-to-face and online. One participant talked about creating a community using an online platform:

> I started the sex workers support group and that has been a really great community. It's mostly a [social media] group where we can kind of just share things, like I posted this there and told them to sign up. And, so, yeah, it's really nice to be able to ask someone like 'have you seen this guy before' and be like 'oh yeah, he likes 'em young, oh yeah he's . . . don't see him.'

Another participant explained how a sex worker run peer group made a valuable difference to her sense of community:

> I've heard so often, like, 'I don't know where I would be if I didn't have someone to talk to that understands' so, you know, to be able to have someone to message at two o'clock in the morning or a call that understands and has been there and done the same kind of work. That makes a big difference.

Within these support circles, finding people who share similar views about the work was key.

While creating community with other sex workers was deemed important, not all sex workers had a community of people to lean on, leaving some isolated. For instance, one participant shared a photograph that captured different shadows to represent her experience with outdoor work at night, saying: "That's the shadow of me. Alone, sad, [silence]. It's very lonely." She continued to talk about isolation and the struggles of connecting: "It's not so much not fitting in but just not, just being alone, like. How alone it is."

*Shared Safety.* Shared safety contributes to the dynamics of community care, with responsibility shared across a network. Due to current sex work laws in Canada, limited options exist to access or create safe workplaces. Sex workers hold the unjust responsibility for their own safety. One participant who worked outdoors explained: "I carry a pocketknife. Besides that, I've taken self-defense courses and things like that." Some participants described times when they knew the situation was unsafe, but they had no one to call and needed the money. One participant shared a time when they had no source of income, no support, had been turned away by services, and had no place to sleep. They shared photographs of the night sky—dark places, where they would hold their breath and did not know "If I would come out alive."

For many participants in this study, ensuring individual safety often meant having a network of people they could reach out to. Individual safety was created by a collective safety net. Sharing a safety plan with community members was a common safeguard against violence: telling people where they would be or if they were in a threatening or dangerous situation and having someone they could call. One person described that "To keep myself safe, I'll basically tell a friend I'm going out and then tell them if I'm not back by a certain hour call my phone. And if I don't answer, just like try and locate me." Safety planning was a critical aspect of the work, as this participant noted:

> Like you gotta have somebody know, like just in case, or if there is any way you could have security, or get to know the person first, just to be sure. 'Cause it's, scary. Like the sex worker died here not long ago, was murdered by an out-call, so, you never know.

Other community safety measures participants discussed included working together, working in a co-operative space, and never working alone. Cooperative arrangements offered a level of control with screenings that promoted safety. One participant explained that a safe work environment was a key motivator for creating a co-op: "We just needed a safe place where we could, you know, entertain people." When working from home, having another sex worker in the house/apartment was a strategy for safety: "There is always a male sex worker home when I have clients. And when he has clients, I'm home."

Shared safety emerged as a strong aspect of community care expressed in multiple ways by study participants. Sex workers looked for ways to ensure their own safety through sharing their work location with reliable people in their circle. Some chose to work in shared spaces for the safety it provided, as well exercising unwritten rules of conduct and outwardly expressing protective methods for themselves and their fellow sex workers. Other sex workers have spoken out about the need for alternative means of safety outside of police protection, as police and public stigma impacted their ability to work safely and to combat the interpersonal violence committed against them. Outdoor work and out-calls presented more safety risks compared to indoor work. One participant described "unsaid rules" sex workers maintained to keep one another safe during outdoor work. They would watch out for one another, because as one person commented "We knew no one else had our back except us. Who is going to really care for the 'lowest of the low' [uses air quotes], than the lowest of the low?"

*Programs and Services.* In the province where this research was situated, two main, urban-based programs for sex workers exist. One program, "Program A," is publicly positioned as a program for individuals wanting to exit the sex industry and is open to

people 30 years and younger across all genders and sexualities. Participants expressed different opinions about this program's requirement that people who want to access services must be trying to leave or "exit" sex work. Participants shared that these services were often framed with encouragement and an orientation on leaving sex work, although individuals accessed the programming from different positions:

> There's some girls, like myself, who are still in the industry, and there are some that are still kind of there because they have no choice, you know, to feed their kids or whatever and there are other women who are out completely but still go there for, like, therapy.

The second program, "Program B," is positioned as meeting women and gender diverse people in the sex industry in terms of where they are at in their lives. Program A received significantly more funding than Program B; however, a significant barrier to receiving support from Program A was the waitlist for services: "There are upwards of 17 of us, last I checked. There's over 50 people on a waitlist trying to get in. Right, but they can't help everyone."

Participants described the respective program they accessed as supportive resources for the sex work industry. Services in both programs were described by participants as places they could go and be themselves and find community. Participants identified that the services offered by both programs were valuable.

> Because there's [Program B] and then there's [Program A], both are separate things . . . . [Program A] has the case management support to help those exit sex work, while [Program B] has the community support aspect, and having both of those services under one roof would be ideal.

When describing both programs, participants shared that staff would go beyond the traditional workday to provide support. Many participants mentioned that staff at Program B were exceptional in supporting their needs. One participant shared a story about dealing with a violent ex-partner, and Program B staff provided unconditional support:

> . . . She was there and talked about, "Do you want to report it, and what would that look like if we did? We don't have to. Would you feel comfortable with an emergency protection order?" Or like, "What would make you feel safe right now?" And she covered everything, right? . . . Everything. They are the best support. If they were everywhere that would be awesome.

Flexible and dedicated staff support individualized to participants' unique needs was described as crucial to the programming and support of both programs.

Many participants commented that more programs and services were needed in the province for male, non-binary, and transgender sex workers to make services accessible to "people of all marginalized genders." Geographical location was also a crucial consideration. At the time of this study, minimal resources were available in rural and small communities:

> There's nothing out there, as far as I've seen . . . Just because where it is such a small community, nobody thinks that people out there are doing it but like I always tell everyone, there's a lot more of us than you would think.

When it came to rural areas and small communities, participants found privacy was a critical component in service delivery. Having a program that supported sex workers must be part of a larger organization to ensure anonymity:

> If you do access a service, then someone's gonna go see you access the service. Right? If you walk into some building in a rural community of like 500 people, some lady's gonna be watching you out her window. And if she knows who you are she might call your mother, right?

Many participants noted that a collective space accessible by surrounding communities was more feasible than having satellite programs in each small community.

Online and phone support services were important for sex workers to access information. Examples included a 24/7 phone line created for and by sex workers, a partnership with the province's Sexual Assault Centre to report information about bad clients anonymously. A mixture of in-person and online/remote access services was deemed essential by participants, particularly those in rural regions.

Participants characterized all the programs they used as having a positive impact on their lives. One participant shared that: "The more supports that you have, the better off it seems. 'Cause, like I was, I was a hot mess, I will admit it, before I joined this." Programming provided support when it came to community care and had an impact on the daily lives of sex workers.

## 4. Collective Care

Collective care includes policies, legislation, and structures at a systems level. While participants expressed different views about whether sex work should be decriminalized or legalized, everyone made it clear that the current systems and policies fail to protect and support people engaged in sex work. Participants expressed that everyone has the right to be safe in the work that they perform. What was most evident from this research was that when laws and policies were enacted based on moral positions that disregarded the vast experiences and motivations of individuals in the field, it had detrimental and harmful impacts on the lives of sex workers and the lives of others. The government plays a critical role in population health and safety for all citizens. This includes sex work legislation and policies and ensuring that these are effectively translated into practice. Described below are the three themes that encompass collective care: (1) laws and policies, (2) accessing services and resources, and (3) attitudes and education.

*Laws and Policies.* Participants expressed the need for government entities to work alongside front-line organizations and not solely with organizations that align with current government perspectives. Without this engagement, the government risks ignoring the diversity in the field. Current concerns include confusing laws, the legal conflation of sex work with sex trafficking, and identifying the ideal legal model.

Many participants noted that Canadian laws are confusing and lack clarity. Individuals have no place to go to receive a direct answer on what the current laws actually mean, which creates fear:

> [I] didn't know whether it was legal, illegal, you know; what the boundaries were cause like I heard from some girls that it was legal to sell it but not legal to buy it . . . So I was kind of not entirely sure how to go about that, um, but where [Name of Street] is so close to the police station, um, I always had that fear, whenever I heard sirens, that they were coming for us.

Current Canadian laws are described as the Nordic Model. How this law is operationalized fails to support its intention, particularly when it comes to addressing systemic issues and safety. One participant shared: "There is a real lack of access to a lot of things that would really help people leave the industry . . . . free education, childcare, healthcare, um, access to housing." The Nordic Model, as it is set up in Canada, fails to address these structural gaps.

Current laws also create issues concerning taxation. Trying to legitimately file taxes was problematic for many sex workers and confusion around laws impeded participants' access to accounting services: "Even just to sit down with an accountant, I had like one person I met with who straight-up told me what I was doing was illegal and, "You're a criminal and you could go to jail." Another participant described a story of hiring an accountant and a financial advisor to help ensure that they were not breaking the law:

> I have like a business and it needs to be, you know, legit. I don't wanna be just you know, I need to have things on paper. I need to be real . . . I have this money, I hired an accountant, I had a financial advisor, I want to pay taxes. Just tell me how to do it the right way.

Participants also shared that current Canadian laws do not effectively support safe working conditions. Several people explained how they needed to work together in a shared space to keep safe in their work. Yet, in doing so, they broke the law. One participant described the following:

> Safe working conditions is an issue when you're working on your own, even when you're working on the street. Part of the Canadian law is that you're not allowed to materially benefit from someone else's work. So that means if my friend and I are working together we have a brothel, and that is a really big issue.

The lack of legal clarity is one way in which the current laws fail to support sex workers.

Participants also held different opinions on whether legalization or decriminalization should be the model in Canada. One participant shared that legalization would help with regulation and ensuring safer work environments:

> I feel that legalizing sex work is the only way, because if sex work was legalized they could regulate it. They could tax it if that's what they really wanted. But for the most part, women should have somewhere, men should have somewhere safe that they feel they are able to go. There should be regular STD checks because every time you go see a sex worker it's a risk you're taking, as well as the sex workers themselves taking this risk . . . It should be safe, and clean, and consensual.

Some participants said that legalization could provide regulations that specifically aim to protect sex workers. One participant explained, "It should be made to protect the girls . . . Because many have been hurt, killed, and . . . they need some kind of, type of security. In every aspect and in every way, and some kind of law got to, or needs to recognize that."

A number of other participants advocated for decriminalization over legalization, such as this individual who explained the following:

> Decriminalization takes away all of the [laws] . . . so you can do sex work. And there's no conditions put in place . . . Because a lot of people don't understand the difference and so you'll hear well-meaning people saying, "We should really legalize sex work." And I'm going, "No, we shouldn't." Us sex workers don't want that, we don't want it to be legalized. That just makes it, it's more problematic because we could still be charged for things because there's still laws about sex work.

While participants did not express consensus surrounding decriminalization or legalization, they wanted the current laws to change. Canadian laws require revisions to better address sex worker safety, in part, by including core aspects that support choice of location and work colleagues. Participant remarks made clear that legislative challenges lie in distinguishing trafficking (violence) and sex work to respect and support the autonomy and needs of people who work in this field by choice.

*Accessing Services and Resources*. Participants identified the services and resources that would best support sex workers. They also talked about the barriers and facilitators related to accessing specific necessities such as basic needs, housing, healthcare, child services, and police services. For instance, one participant shared a picture of a bench to represent her housing: "I mean it served as like a bed for me for a while and stuff so I wasn't sleeping on the cold ground." Another participant also spoke about homelessness in sex work: "A lot of people who are involved, you know, they stay on the streets, they don't really have a place of their own." Shelters and secondary housing were accessed by a number of participants at some point in their lives, who described several issues with the current shelter system: "It was a safe place but I didn't, I didn't get a lot of support from them and I still run into women that have been there, when I was there, and they have a bad taste for that place too." Comments highlighted that shelters may not be as responsive to the needs of sex workers and fall short of their mandate to support those who stay in the facilities.

Access to and the delivery of healthcare services were identified as challenging for many people involved in sex work. Some participants shared how they experienced judgement from healthcare providers:

> I had a doctor who I just never felt comfortable enough to say, 'Hey, this is what I do, can I get, like, monthly testing or whatever?' So, it's definitely hard for people who don't have access to good doctors to, to feel, I guess, safe enough, not judged, to say, 'Hey, I'm a sex worker can I get monthly testing or whatever?'

Some people spoke about their experience of being refused access to care at hospitals. One participant admitted that they did not disclose their work to ensure they received the necessary healthcare: "I've lied through my teeth many times to make sure that I don't have issues accessing healthcare."

Counselling and therapy were identified as an important aspect of healthcare. Participants described the impact and the power that counsellors had in their lives, including support and connection. Having counsellors who understood the different contexts of sex work was critical, and this can permit the discussion of work with openness without being blamed or pressured to exit, as this person expressed: "One time that I did mention sex work to a psychiatrist I was shut down completely and they wanted to hear no part of it." Another participant made the following comment:

> And so, you're not going to go to a therapist or a psychiatrist, the one prescribing you medication, and say that, "I'm having a bad day. And it's not because I don't enjoy sex work but because these experiences have happened to me this week and it's ... " They don't want to hear it because you've put yourself in that situation.

Participants shared that the healthcare system, including therapy contexts, was not easily accessible for sex workers due to judgement and stigma from healthcare personnel.

Several participants also noted that Child Protective Services (CPS, also known as child welfare) took issue with their work. Individuals shared several inappropriate comments directed at them by personnel as well as painful experiences that they had, such as the following:

> One of the last meetings I had with my worker, she looked at me and said, 'Just because you look good on paper doesn't make you a good parent.' Oh really? But it's kind of funny when the public health person doesn't have an issue with me ... [Name] works with Child Services doesn't have a problem with me, says I'm doing an awesome job, and for you—as a child worker—to come into my home or anybody else's home ... and say [that].

Ignorance, comments laden with moral judgements, and a lack of understanding of policies and laws within the federal and provincial government systems foster distrust. Experiences sex workers had with various services repeatedly highlight that their profession was a fundamental issue for the system. As one individual expressed, "I'm supposed to have faith in the police and the system and child services? Ha-ha-ha. Go the fuck away. How can you expect me to trust a system that's been broken, still is broken to this day, and be okay?"

When it came to the police, participants had varied interactions. One participant shared a positive experience when the police came into an adult massage studio to address threatening phone calls: "We had the police come in one time because we were getting like threatening calls towards the studio and stuff and the police were great. They came and talked to us and they were good." Another individual also shared her helpful experience interacting with an officer who did not judge her or her work. However, several participants also shared stories of not going to the police due to fear, history of past negative experiences with the police force, and feeling judged:

> I didn't go to the police because of, I was an escort and the stigma [of] that, or the role that police play in our, in our life is not a positive one. I didn't need to go to

the police station and be judged for being an escort, you know? To be judged for finding food to put on my children's plate, you know?

Similarly, another participant shared that turning to police for protection from violence or follow-up on an assault was complicated:

If you get beaten up at work you are less likely to go to the police and say, 'Hey, these are the details surrounding me being assaulted.' If you were assaulted at any other type of job, you would have no fear [of] going to the police station right away. But when it comes to sex work, it's something that you have to think over in your head. You know, am I going to get in trouble if I go to a police officer and say I've been assaulted and this is what happened. Or another big one is, are they going to believe that I was raped if I'm a sex worker? Which is a huge issue, because chances are if you had sex with someone against your will, they're not going to believe you were sexually assaulted. You're a sex worker. It sounds like you're asking for it, but you're not.

Location also played a factor in sex workers' experiences with police. In smaller towns and communities, reaching out to the police was rarely an option due to community exposure. Privacy could easily be compromised because friends or family in the community may be in a role that provides access to privileged information.

*Stigma, Attitudes, and Education.* Stigma was a central aspect of many participants' experiences within systems and services. One individual commented on ongoing stigma, saying:

These are people too. These are women. These are somebody's daughters, somebody's mothers, somebody's sisters that . . . [it] doesn't matter what kind of background you come from—because I'm a mom, coming from a bad home . . . Something's got to be put in place to protect these girls as well, because someone's, somebody's been hurt or killed because an asshole wanted to, "Well I can shoot a whore" for lack of a better word or lack of a better term. "She's a whore so she's just trash."

Stigma prevented sex workers from accessing services:

There's a fear of telling people you're a sex worker, right? And so, for a lot of sex workers, it's not that they don't have access to certain things, but it's that they are too embarrassed to seek them out, or they're too ashamed to seek them out.

Changing attitudes and educating the public was deemed critical to improving the everyday lives of sex workers in this study. Efforts that target service providers were desperately needed to support collective care. Local events are needed to bring awareness to sex work, including marches, protests, festivals, and panels. Mitigating negative attitudes and stereotypes that feed stigma takes time and legislative changes. Well-funded services and better public education hold considerable potential to improve this landscape.

## 5. Limitations and Considerations

This study has limitations worth noting. While researchers were able to interview nonbinary persons and people who self-identified as women, the final study sample, according to what people chose to share, lacked greater diversity with respect to disabilities, race, gender, and sexual orientation. This research study was also conducted in one province. Future studies are needed across Canadian regions to develop a robust care-based framework possessing comprehensive structural interventions that consider the voices of Canadian sex workers.

Methodologically, there are also several areas worth noting. Participant's disparate positions regarding geographical locations, desire for anonymity, and transitions with work and life presented unexpected challenges in striving to meet the needs for all participants. While the photo-elicitation interviews created a space for both individual narrative, and group engagement presented opportunities for rich discussion, the group-based format

was not feasible or wanted by several participants. Thus, while creating a blended option for engagement was thought to be ideal, admittedly, there is a need to further consider the implications of power, voice, and group-dynamics when conducting research in this area. While the research team was saturated in the visual and textual data of the photo-elicitation interviews, however, the opportunity for participants to provide further feedback on the second phase of findings could offer additional insights and clarity. While requiring this level and longevity of engagement can be onerous for the participants and difficult to foster and maintain, it is an area that requires further consideration. Questions surrounding power-sharing, format, and voice linger and also remain areas for further methodological exploration, particularly when working with structurally vulnerable populations whose voices have often been silenced by institutions and academia.

## 6. Conclusions

To be effective, structural interventions must acknowledge the inherent diversity in sex work, centering the voices and experiences of sex workers in the development of policies and systems to support the needs of all persons. Building capacity for collective care requires accountability. Addressing areas of care is crucial for improving the quality of life for people engaged in sex work. Emerging from these findings is an accountability care framework consisting of three primary components:

1. Creating a liaison/task force between people in the sex work industry and municipal regions and provincial government;
2. Developing guidelines and best practices across identified areas of collective care; and
3. Creating education and training sessions led by and in consultation with sex workers.

These recommendations towards accountability care are rooted in the principle that sex workers will take the lead in guiding the recommended processes to develop the necessary support. This, in turn, depends on implementing the following measures guided by meaningful participation and consultation with current sex workers.

### 6.1. Task Force Liaison

Sex workers need a seat at the table with respect to micro- and macro-level decisions that affect them. A task force would ideally encompass people who (a) currently engage in sex work in the area, (b) have experience working in the sex industry, and (c) live and/or work in the region. Activities and decision-making would focus on federal and provincial funding development and reviews, regulations and policy development, and legislation across municipal, provincial, and federal levels. All roles would be paid positions, with mechanisms in place to include people who require anonymity. Recruitment for this task force must seek a diversity of experience and expertise across the spectrum, acknowledging that this is not simple. This requires patience from community members to engage in this process, especially if individuals have prior negative experiences with authority or government entities.

Looking at other national and international examples of such groups is helpful. For instance, in Sydney, Australia, a sex industry liaison is in place between the industry and the municipality (paid for by the government) to build open communication and trust that has proven to be effective (Jeffrey 2018). This role has worked on location regulations in the areas of health, safety, and good working conditions (Jeffrey 2018). While the political context differs from Canada, such a model demonstrates how a taskforce at the municipal and provincial levels could successfully guide regulations regarding health, laws, and resources.

### 6.2. Guidelines and Best Practices

Recommendations for guidelines and best practices to improve the quality of life for sex workers must address self-care, community care, and collective care. They should also focus on (a) changes to Canadian laws at municipal, provincial, and national levels, and (b) additional supports and resources, primarily for open-ended supports that do not require

individuals to leave sex work. All study participants agreed that current Canadian laws and how they are operationalized are not working, pushing sex workers further underground and making working conditions more unsafe. Violence and safety continue to be key issues, and participants seek more control over who they work with, how they work, and where they work.

Current laws support the stigmatization and criminalization of sex workers, which directly increases violence against sex workers. Canada's current *Protection of Communities and Exploited Persons Act* (Department of Justice 2014) is a barrier to accessing services and advocating for policy change to make services more accessible. Safe working conditions are a critical issue requiring legislation. Several participants expressed that violence and safety are key concerns for them. Participants want more control over all aspects of their workplaces, speaking to a need for labour protections within the industry. Policies must be changed to let sex workers work together and in different settings (online or face-to-face) to increase safety. Independent entrepreneurs working indoors includes individuals working together in a co-operative format or working alongside another person for safety; however, these workers are violating current Canadian laws by working collaboratively. This is highly problematic because sharing workspaces or co-operatives provides sex workers with more control over their workspaces, while providing them with support in screening, security, and safety. Safe work options supported by legislation are needed so that sex workers can work where they need and with whom they need to be safe.

Safe work options supported by municipal, provincial, and federal legislation are essential for sex workers. This can be achieved by decriminalizing sex work. It can also be actualized by legalizing shared workspaces and co-operatives (e.g., shared residential in-call spaces) and allowing sex workers to hire support staff for screening or security.

Beyond recognizing the myriad of experiences across the continuum of sex work, service providers for people in the sex industry must be willing to shift away from meeting complex situations with blanket solutions that include exit-only programs or supports designated for trafficking survivors. One participant articulated this well by saying: " . . . everybody needs something different and wants something different and needs a different outcome." The following specific recommendations are identified as urgent action items:

- Inclusive supports for all genders;
- Access to affordable housing and shelter;
- Improving access to food, clothing, no-wait counselling, the Internet, and cell phones;
- Uncomplicated healthcare access and universal dental and optical insurance;
- Rural supports for sex workers linked to broader programs so people can access them confidentially or anonymously;
- A central resource hub accessible in person or online, with 24 h support offering information on sexual and mental health topics, marketing, managing clients, and balancing finances;
- Creating policies and bylaws that affect the work safety for sex work is needed.

*6.3. Ongoing Education and Training Led by and in Consultation with Sex Workers*

Given the diversity of experiences in sex work, developing appropriate curricula for service providers across a variety of roles and professions is essential. Training sessions should be delivered by skilled facilitators with expertise on sex workers' diverse experiences. Target audiences include current and future healthcare practitioners (physicians, nurses, social workers, and psychologists), as well as law enforcement, frontline staff, case workers, accountants, and any other profession that may provide valuable services to sex workers. Curriculum developed by and with sex workers will raise awareness and address stigma and judgement, which were heavily cited as deterrents for connecting with professionals. Non-judgmental support, free from a focus on 'exiting', requires the following: meeting people where they are at in their present circumstances and beliefs, demonstrating compassion, and showing humanity. Service providers must recognize their own biases that potentially impede services and supports.

A critical part of this curriculum is understanding sex work as existing on a spectrum of choice, which is paramount to providing adequate, appropriate care. Recognizing that the spectrum of sex work is critical, and at times—similarly in any job—exploitation occurs. People working in the sex industry can find themselves in exploitative situations based on power and their ability to make a decision for themselves. At other times, the sex industry can be empowering. Contexts and experiences dramatically differ, and it is important to always consider the interplay between control, power, security, and autonomy across the sex work spectrum.

**Author Contributions:** Conceptualization, K.C.S.; data curation, M.R.P.; formal analysis, K.C.S., A.G. and M.R.P.; funding acquisition, K.C.S.; investigation, K.C.S.; methodology, K.C.S. and A.G.; project administration, K.C.S. and H.J.; visualization, A.G., M.R.P. and H.J.; writing—original draft, K.C.S.; writing—review & editing, K.C.S., A.G., M.R.P. and H.J. All authors have read and agreed to the published version of the manuscript.

**Funding:** Social Sciences and Humanities Research Council of Canada (SSHRC IDG 430-2016-0058).

**Institutional Review Board Statement:** The study was approved by the Institutional Review Board of University of Calgary (REB17-1503).

**Informed Consent Statement:** Informed consent was obtained from all subjects involved in the study.

**Acknowledgments:** We would like to thank the participants in the study. We would also like to thank Haley Anderson, Amy Burke, Janice Kennedy, Paula Sheppard, Laura Winters, and Jenny Wright. We would also like to thank the reviewers for their thoughtful feedback.

**Conflicts of Interest:** The authors declare no conflict of interest.

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
