# Peer review of "“We Knew No One Else Had Our Back except Us”: Recommendations for Creating an Accountability Care Framework with Sex Workers in Eastern Canada"

_socsci, doi:10.3390/socsci11080366_

Round 1

Reviewer 1 Report

An excellent paper.  Well written and a very useful framework used to identify key issues.  The recommendations are very worthy. 

Only two minor issues.

Can the literature review be more extensive around discussing the key issues emerging around your research questions, and why this study is needed to address the gaps.

2.  If you had to redo the study what would you do differently?  What methodological improvements would you suggest?

Author Response

Thank you for your feedback. We have addressed the following:

Comment 1: Can the literature review be more extensive around discussing the key issues emerging around your research questions, and why this study is needed to address the gaps.

Thank yo for the suggestion. In response, we have expanded the introduction and literature on pages 2-5, adding seven additional pieces of scholarship surrounding our research (both on the necessity of the research as well as the theoretical framings concerning care). Additional sections include the following:

Sex worker advocacy groups advocated for timely and long-term government interventions, including: no-qualifying requirements for social supports, healthcare, and a universal income; ensuring sex workers have access to essential resources such as condoms and personal protective equipment; and access to safe and affordable housing (Shareck et al. 2021). These types of interventions require an understanding that uni-sector policies fail to address the overlapping identities and challenges commonly experienced by Canadian sex workers (Shareck et al. 2021). The ongoing COVID-19 pandemic has highlighted sex workers’ need for multi-sectoral policy that crosses the domains of income, gender, housing, healthcare, and disability.

Sex workers exist in marginalized social, political, and economic positions (Zangger 2010). Government and institutions abdicate care for sex workers, perpetuating structural violence and stigma. Sex workers must provide care for themselves and their communities from the margins of society. (P3)

and 

. Emerging evidence suggests care in the form of peer support is a strong mechanism for resisting sex work stigma. Huang’s (2016) research with Australian sex workers identified peer support as a key mechanism for shifting sex work stigma and its deleterious impacts. As Ahmed (2014) writes concerning sex care in relationship to structural violence: “Some have to look after themselves because they are not looked after: their being is not cared for, supported, protected” (np). (P4)

2.  If you had to redo the study what would you do differently?  What methodological improvements would you suggest?

Thank you again for this thoughtful question. In response we have added the following reflection under "limitations":

Limitations and Considerations

This study has limitations worth noting. While researchers were able to interview nonbinary persons and people who self-identified as women, the final study sample, according to what people chose to share, lacked greater diversity with respect to disabilities, race, gender, and sexual orientation. This research study was also conducted in one province. Future studies are needed across Canadian regions to develop a robust care-based framework possessing comprehensive structural interventions that centre the voices of Canadian sex workers.

Methodologically, there are also several areas worth noting. Participant’s disparate positions regarding geographical locations, desire for anonymity, and transitions with work and life presented unexpected challenges in striving to meet the needs for all participants. While the photo-elicitation interviews created a space for both individual narrative, and group engagement presented opportunities for rich discussion, the group-based format was not feasible or wanted by several participants. Thus, while creating a blended option for engagement was thought to be ideal, admittedly, there is a need to further consider the implications of power, voice, and group-dynamics when conducting research in this area. While the research team was saturated in the visual and textual data of the photo-elicitation interviews, however, the opportunity for participants to provide further feedback on the second phase of findings could offer additional insights and clarity. While requiring this level and longevity of engagement can be onerous for the participants and difficult to foster and maintain, it is an area that requires further consideration. Questions surrounding power-sharing, format, and voice linger and also remain areas for further methodological exploration, particularly when working with structurally vulnerable populations whose voices have often been silenced by institutions and academia.

Reviewer 2 Report

The paper focuses on sex work as a social phenomenon explored from the perspective of the theory of intersectionality. It aims to highlight the role of support of sex workers as being imbedded in the notions of ‘care’ understood as self-care, community and collective care. By focusing on these levels, the authors apply a holistic view of care and support which allows them to highlight in more details the diversity of the lived experiences and identities of sex workers. The empirical part of the study is based on an interesting methodological approach of photo-elicitation interviews. It includes an analysis of photos of the lived experiences taken by the participants in the study and interviews with them. Overall, it is very well written paper which clearly presents the design of the study and the empirical findings and I have a few suggestions to the authors. It would be informative for the readers to read more about the specifics of the critical framing approach that the authors used for the data analysis. It would be also very informative to know more about the way the photos and the narratives on them were analyzed. What are benefits of this approach and what are the main findings from the analysis of the visual material?  

Author Response

Thank you for your feedback. We have addressed the following:

Q1. It would be informative for the readers to read more about the specifics of the critical framing approach that the authors used for the data analysis. It would be also very informative to know more about the way the photos and the narratives on them were analyzed. What are benefits of this approach and what are the main findings from the analysis of the visual material?  

Response:

Thank you for your comments and feedback. The critical framing approach guided the group process. We have written about this elsewhere, and we have now included that citation in the article. However, this focuses on revisiting the interview data and guided by the interview questions. Critical framing did not take a central role in this process. We recognize that the way this portion of the manuscript is written does not make this clear. We have reworked this section, as well as the abstract so that critical framing is noted but associated with group process.  We have also added further details about the visual material:

Participants were asked to take photographs of their life experiences as a person who engages in sex work, followed by an individual interview. Interviews took place during 2018-2019, and were conducted face-to-face at the community partnership organization, at a program location, or a public location chosen by the participant. Participants had the option of participating in two follow-up group meetings. In line with best practices, all participants received honourariums for the interviews and group meetings.

            Preliminary Analysis and emergent themes that guided the group process involved critical framing (Sitter, 2015). This process recognizes that a participant’s interpretation of their image carries the most power and importance of visual meaning (Stanczak 2008). When large amounts of data are present, researchers undertake analysis within the context of the other data and the overall theoretical framework (Guillemin and Drew 2010, 184). The preliminary themes were shared at the group meetings for feedback and discussion. The final group themes from this process included: basic needs, laws, healthcare, the industry, and supports.

While key learnings from the group sessions have been documented elsewhere (see Sitter et al. 2020), this article provides a focused review on the photo-elicitation interview data. As part of the interview, participants used photographs to convey and represent experiences that affect their lives as a person living and/or working as a sex worker. Interview questions were based on image guidance, specifically how the images represented 1) life experiences as a person who engaged in sex work; 2) social and personal factors; 3) laws; 4) geography (work, travel, home); and 5) needed supports. As a second form of analysis focused on the interview data, members of the research team revisited and reviewed all transcripts for a period of four months. Guided by the research questions, all interview data were reviewed and involved sustained engagement until clear themes emerged. This included the ways in which participants described their photographs in regards to providing further context to their experiences. We noticed the theme “supports” was linked to each theme named above, providing a deep structure that framed the data. Through an in-depth analysis of the individual interviews, the idea of supports evolved to encompass aspects of care, which are used to organize the findings and recommendations. These four aspects are broadly defined as follows:…